# Validation of the Italian version of the Dark Tetrad at Work scale

**Francesco Marcatto**[ID][1]*, **Lisa Di Blas**[1], **Donatella Ferrante**[ID][1], **Isaiah Hipel**[2], **Kevin Kelloway**[2]

**1** Department of Life Sciences, University of Trieste, Trieste, Italy, **2** Department of Psychology, Saint Mary's University, Halifax, Canada

* fmarcatto@units.it

**Data Availability Statement:** All data files are available from the OSF database: https://osf.io/8mj73/?view_only=fa1f7ac647e54e63b6103ea3aa151ebf.

## Abstract

This study presents and validates the Italian adaptation of the Dark Tetrad at Work (DTW) scale, an instrument for assessing four socially aversive personality traits (narcissism, Machiavellianism, psychopathy and sadism) in the context of the workplace. A total of 300 Italian-speaking participants (50% female, M age = 32 years ± 9.2) and 253 English-speaking participants (38% female, M age = 39 years ± 12.1) were recruited via an online survey platform. The Italian-speaking sample was used to test the factorial structure, reliability and criterion-related validity of the Italian version of the DTW, whereas the English-speaking sample was used to test cross-language measurement invariance. Results from confirmatory factor analysis showed that the original four-factor model provided the best fit to the data. The Italian DTW scale demonstrated acceptable internal consistency, with reliability coefficients of $\omega$ = .77 for narcissism, $\omega$ = .80 for Machiavellianism, and $\omega$ = .81 for both psychopathy and sadism. Concurrent associations between the DTW scales and negative and positive workplace outcomes supported the criterion validity of the scale. Machiavellianism, psychopathy, and sadism positively correlated with counterproductive work behaviors and workplace bullying, and negatively with organizational citizenship behaviors and affective organizational commitment. In contrast, narcissism exhibited a unique pattern: It correlated positively with positive workplace behaviors and negatively with counterproductive behaviors toward the organization, but it was also found to be a significant predictor of workplace bullying. This finding may reflect multidimensional nature of narcissism, but a note of caution is warranted in interpreting this result, as all measurements relied on self-report instruments, introducing the possibility of socially desirable associations influencing the outcomes. Finally, the comparison with the English sample established configural, full metric and partial scale invariance, allowing for valid cross-language comparisons between Italian and English-speaking populations in the future. Preliminary Italian normative data were provided to offer a benchmark for the interpretation of DTW values. This study provides a reliable and valid instrument tailored to the Italian workforce, enhancing our understanding of dark personality traits within organizational contexts and providing organizations with an effective means to address and manage dark personality traits for a healthier workplace culture.

**Funding:** The author(s) received no specific funding for this work.

**Competing interests:** The authors have declared that no competing interests exist.

## Introduction

In recent years, research in organizational psychology has increasingly focused on exploring the influence of personality traits on workplace behaviors, interactions, and outcomes. Among the various personality frameworks, the Dark Tetrad has emerged as an important area of study. It comprises four socially aversive personality traits associated with various maladaptive behaviors and interpersonal difficulties [1]. The four traits that make up the Dark Tetrad are (subclinical) narcissism, Machiavellianism, (subclinical) psychopathy, and (subclinical) sadism.

Narcissism is characterized by a grandiose self-view, fantasies of control, success, and admiration, and a sense of entitlement [2]. Individuals high in subclinical narcissism often exaggerate their accomplishments, block criticism, and refuse to compromise [3]. The trait Machiavellianism is named after the Italian Renaissance diplomat and writer Niccolò Machiavelli. Individuals with high levels of Machiavellianism are characterized by manipulation, deception, and a cynical view of human nature [4]. They are strategic and calculating, exploiting others for personal gain without empathy or concern for their well-being [5]. Psychopathy is a personality disorder characterized by a lack of empathy, shallow emotions, and a disregard for the rights and feelings of others [6]. Individuals with high levels of subclinical psychopathy do not exhibit the unstable antisocial lifestyle characteristics of clinically defined psychopaths [7], but they still tend to be impulsive and lack social regulatory mechanisms, guilt, and remorse [8]. Subclinical sadism can be viewed as the enjoyment of cruelty in many everyday activities, such as bullying or intentionally humiliating another person (direct sadism), and taking pleasure in watching violent films or hurting people in video games (vicarious sadism) [9].

The construct of sadism was met with skepticism in the early Dark Tetrad literature: its similarity to psychopathy, as both traits involve a proneness to aggression [10], raised doubts on its unique contribution beyond the established traits of the original Dark Triad (narcissism, Machiavellianism, and psychopathy) [11]. Despite possible overlaps with psychopathy, the enjoyment of cruelty is a unique feature of everyday sadism and distinguishes it from other antisocial traits [12]. Indeed, extensive research has shown that everyday sadism consistently predicts various deviant behaviors and outcomes beyond psychopathy and the other socially aversive personality traits [11, 13, 14].

### The Dark Tetrad in the workplace

The effects of the Dark Tetrad traits can be profound and harmful, both to the individuals who exhibit these traits and to the people with whom they interact [15, 16]. In the workplace, the presence of individuals with Dark Tetrad personality traits can have several negative consequences. People with high levels of Machiavellianism and narcissism may engage in manipulative tactics to achieve their goals, often at the expense of others, and their lack of concern for others well-being may lead to exploitative and harmful behaviors [17]. For example, Machiavellianism has been identified as a significant predictor of employees' willingness to engage in corruptive behavior [18]. Similarly, narcissism has been found to be a positive predictor of the adoption of bullying tactics at the workplace [19]. High levels of psychopathy are associated with antisocial behaviors such as impulsivity, aggression, and disregard for social norms and rules. This can lead to legal and interpersonal problems, as well as difficulties in adhering to social and occupational norms [3]. In fact, people with high levels of subclinical psychopathy are more likely to intentionally engage in behaviors aimed at harming their organization [18, 20]. Lastly, people with high levels of subclinical sadism derive pleasure from others' suffering

and are therefore prone to workplace bullying, such as shaming others and openly attacking their dignity and self-esteem, especially in a public context [19].

In summary, the manipulative, exploitative, and antagonistic behaviors associated with the Dark Tetrad traits can create a toxic work environment and negatively impact the overall productivity, morale, and employee well-being [21].

## The Dark Tetrad at Work scale

Due to the relevance of the Dark Tetrad, several scales have been developed to assess Dark Tetrad traits in different contexts, including interpersonal relationships and online behaviors. The Dark Tetrad at Work (DTW) scale was recently developed as a contextualized measure of Dark Tetrad traits specifically targeted to the work environment, where the manifestation of these traits may differ from other social contexts [22].

The DTW scale measures each of the four Dark Tetrad traits with a total of 22 items (four items for Machiavellianism and six items for each of the other traits) and has been shown to possess adequate psychometric properties, including construct validity, test-retest reliability, and predictive validity [22]. In particular, two studies have confirmed the good fit of the hypothesized four-factor structure, the stability over time, and the significant associations of the measured traits with various workplace outcomes, including workplace deviant behavior, bullying, and organizational citizenship behaviors. Further studies have confirmed the incremental power of the DTW scale in predicting counterproductive work behaviors and job performance over the traditional Big Five personality dimensions [23, 24].

To the best of our knowledge, only the original English version and a translated Spanish version [23] of the DTW scale are available. The aim of this article is to report on the development and validation of the translated Italian version of the DTW scale, which includes a comprehensive testing of its psychometric properties and establishing cross-language measurement invariance.

The validation of the Italian version of the DTW scale is of great importance for both research and applied purposes. First, it will provide researchers in organizational psychology with a valid and reliable context-specific instrument to study the prevalence and impact of dark personality traits in the Italian workforce. Second, the cross-language comparisons will provide a more comprehensive understanding of how the Dark Tetrad operates in different work environments and cultural contexts.

## Methods

### Participants

**Italian-speaking sample.** A total of 305 participants were recruited through Prolific, an online survey platform. Compensation and recruitment were handled by Prolific. Eligibility criteria included that participants were at least 18 years old, fluent in Italian, and currently employed. Five participants failed one or both of the attention screening questions and were therefore excluded. This resulted in a final sample size of 300 individuals (50% female) aged 20 to 62 years (M = 32, SD = 9.2).

**English speaking sample.** The English-speaking sample was also recruited through Prolific. Eligibility criteria included a minimum age of 18 years, fluency in English, and current employment. All participants passed the attention screening, resulting in a final sample of 253 individuals (38% female) aged 22 to 85 years (M = 39, SD = 12.1).

## Measures and procedure

The Italian-speaking sample was asked to complete a questionnaire that included the Italian version of the DTW, along with additional organizational measures of workplace deviance and positive workplace behaviors, in order to verify the criterion-related validity of the Italian version. Conversely, the English-speaking sample received a questionnaire presenting only the English version of the DTW, to test its cross-language measurement invariance.

The DTW [22] contains 22 items, six measuring narcissism (e.g., "I am much more valuable than my coworkers"), four measuring Machiavellianism (e.g., "At work, people backstab each other to get ahead"), six measuring psychopathy (e.g., "I don't care if my work behavior hurts others"), and six measuring sadism (e.g., "I love to watch my boss yelling at my coworkers"), rated on a 5-point scale (1 = strongly disagree to 5 = strongly agree).

To ensure equivalence of item meanings between the Italian and English versions of the DTW, a careful translation process was used that included both forward and backward translation. First, the English version was translated into Italian by a bilingual translator. After forward translation was completed, a second bilingual translator (a native English speaker) independently back-translated the DTW into English. The translators then compared this back translation to the original English version. Adjustments were then made to evaluate item-by-item consistency, the clarity, and the accuracy of the translation. The final version of the Italian DTW is shown in S1 Table.

Counterproductive work behavior (CWB) was assessed using the Italian version of the Counterproductive Work Behavior Checklist [25, 26], a 45-item questionnaire that measures the frequency of counterproductive behaviors at work, directed at individuals (CWB-I, e.g., "Refused to help someone at work") and toward the organization (CWB-O, e.g., "Purposely did your work incorrectly"), with answers on a 5-point scale (1 = never to 5 = every day). Reliability scores in the current study were $\omega = .96$ for CWB-I and $\omega = .91$ for CWB-O.

Workplace bullying was assessed using the nine items of the Italian version of the Short Negative Acts Questionnaire (S-NAQ) [27, 28], which was adapted to ask participants to indicate on a 5-point scale (1 = never to 5 = every day) how often they engaged in each behavior in the past six months (e.g., "I have ignored or excluded my colleagues"). The reliability score in the current study was $\omega = .89$.

Affective organizational commitment (AOC) was assessed using the 10-item affective subscale of the Italian version of the Organizational Commitment scale [29, 30] (e.g., "I would be very happy to spend the rest of my career with this organization"), with answers on a 5-point scale (1 = strongly disagree to 5 = strongly agree). The reliability score in the current study was $\omega = .92$.

Organizational citizenship behavior was assessed administering the Italian version of the 24-item questionnaire developed by Podsakoff and colleagues [31, 32], that yields two measures: organizational citizenship behavior directed at individuals (OCB-I, e.g., "I am always ready to give a helping hand to those around me") and directed toward the organization (OCB-O, e.g., "I respect company rules and policies even when no one is watching me"), with responses given on a 5-point scale (1 = strongly disagree to 5 = strongly agree). Reliability scores in the current study were $\omega = .86$ for OCB-I and $\omega = .80$ for OCB-O.

The study was approved by the Ethics Committee of the University of Trieste, Italy, and was conducted in accordance with the Helsinki Declaration. Written informed consent was obtained on the first page of the survey, emphasizing the confidentiality of the data and the participant's right to withdraw from the study at any time. Data was collected between June 30, 2023, and August 20, 2023.

## Statistical analysis

The factorial structure of the Italian version of the DTW was tested using confirmatory factor analysis (CFA) on the Italian sample. We followed the same procedure used in the original validation of the scale [22] and therefore tested the following three models: an unidimensional model in which all 22 items load on a single factor, a three-factor model in which psychopathy and sadism items load on the same factor (i.e., narcissism, Machiavellianism, and psychopathy-sadism), and the hypothesized four-factor model (i.e., narcissism, Machiavellianism, psychopathy, and sadism). Due to the high values of skewness and kurtosis of the data (see S1 Table), CFA was performed using the diagonally weighted least squares estimation method [33]. The following fit indices were considered: CFI (Comparative Fit Index); RMSEA (Root Mean Square Error of Approximation); Tucker-Lewis Index (TLI), and the normed chi-square ($\chi^2$/df). Values higher than .90 for CFI and TLI and lower than .08 for RMSEA indicated an acceptable fit to the data [34, 35]. The normed chi-square was used because it is less sensitive to sample size compared to the traditional chi-square, and values lower than 5 indicated a good fit [36].

McDonald's $\omega$ was used to check the reliability level of the internal consistency [37]. Indices greater than 0.70 are generally considered good indicators of reliability in applied settings [38].

The multi-group confirmatory factor analysis (MGCFA) was subsequently run on both samples (Italian and English) for testing cross-language measurement invariance of the DTW scale. MGCFA consists of a series of progressively restrictive tests of measurement invariance [39]. The first step is the configural invariance, which tests whether the factor structure is the same between groups. The second step, metric invariance (also called weak invariance), tests whether the factor loadings are equivalent across the groups. Lastly, scalar invariance (also called strong invariance) tests whether the item thresholds are equivalent across groups. Configural invariance is achieved when the model fits both linguistic groups, using the same parameters as the CFA. Metric and scalar invariance are evaluated by comparing their models' fit to the previous one, considering $\Delta$CFI $\leq$ -0.010, $\Delta$TLI $\leq$ -0.010, and $\Delta$RMSEA $\geq$ 0.015 serving as the thresholds for accepting measurement invariance [40]. If the difference in fit between the models does not meet the threshold for accepting full invariance, partial metric invariance could still be explored [41].

Finally, criterion-related validity was assessed by calculating the Pearson's correlations among the DTW scales and the other measures of workplace deviance and positive workplace behaviors, and by conducting hierarchical regression analyses with the DTW scales as predictors and the workplace outcomes as dependent variables, controlling for gender and age.

All analyses were conducted using Jamovi software with the Semlj module.

## Results

### Factorial validity: Internal structure and reliability

The results of the CFA on the Italian sample data are reported in Table 1. The unidimensional model showed a poor fit to the data, while both the three- and four-factor models showed a more than acceptable fit to the data, with the latter providing the best fit (chi-square difference test, $p < .001$). The factor loadings of the four-factor model are shown in S1 Table.

Reliability analysis revealed acceptable values. McDonald's $\omega$ was .77 for narcissism, .80 for Machiavellianism, .81 for psychopathy, and .81 for sadism.

A series of analyses of variance (ANOVA) tests were conducted to examine whether the DTW scales scores varied by participant age group and gender. No significant differences in Machiavellianism and sadism scores were found between participants when compared in

**Table 1. Fit indices for the CFA of the Italian version of the DTW scale.**

|  | $\chi^2$/df | CFI | TLI | RMSEA (95%CI) |
|---|---|---|---|---|
| Unidimensional model | 6.17 | .878 | .863 | .134 (.127-.141) |
| Three-factor model | 2.80 | .957 | .951 | .079 (.071-.087) |
| Four-factor model | 2.53 | .965 | .960 | .073 (.065-.081) |

Note. Based on the modification indices, adjustments were made to allow for covariance between errors in all models. Specifically, errors for items 4 and 5, items 9 and 10, and items 12 and 13 were permitted to covary.

terms of age groups and gender (all $p$s > .05), although a significant main effect of gender was found for psychopathy. Specifically, males (M = 1.61, SD = 0.58) had higher scores than females (M = 1.42, SD = 0.59; $F_{(1, 285)}$ = 6.25, $p$ < .05, $\eta^2_p$ = .02). In addition, a significant main effect of age group was found for narcissism ($F_{(3, 287)}$ = 5.42, $p$ < .05, $\eta^2_p$ = .05). Post hoc tests with Bonferroni correction revealed that participants in the 20–29 age group (M = 2.58, SD = 0.68) had significantly lower narcissism scores than participants in the 30–39 age group (M = 2.89, SD = 0.54; $p$ < .001) and participants in the 50+ age group (M = 2.99, SD = 0.77; $p$ < .05).

Descriptive statistics for the DTW scales in the Italian sample are presented in S2 Table.

## Measurement invariance across languages

The first step of the measurement invariance is to evaluate the configural invariance through MGCFA. In this model (M1), no equality constraints were imposed on model parameters across samples. As reported in Table 2, the results of configural invariance showed adequate fit indices, thus configural invariance was established. Next, to assess the metric invariance, the fit of the configural invariance model (M1) was compared to the constraint model where all the factor loadings were constrained to be equal across the two samples (M2). Since the differences of CFI, TLI, and RMSEA were smaller than the cut-off values, full metric invariance was established. Lastly, to assess the scalar invariance, the fit of the metric invariance model (M2) was compared with a model where both factor loadings and all thresholds were constrained to be equal across the two samples (M3). The differences of TLI and RMSEA were smaller than the cut-off values, but the difference of CFI exceeded the suggested threshold, therefore full scalar invariance was rejected and partial scalar invariance was evaluated. According to modification indexes, thresholds 2 and 3 of the third item of the DTW scale were released to be freely estimated (M4) and partial scalar invariance could be established.

## Criterion-related validity

The correlations among the Italian version of the Dark Tetrad and the other measures of organizational outcomes are shown in Table 3. All DTW scales were positively related to each of the measures of workplace deviance (CWB-I, CWB-O, and S-NAQ), except for narcissism,

**Table 2. Fit indices for the MGCFA for testing measurement invariance of the DTW scale across Italian and English languages.**

| Model | $\chi^2$/df | CFI | TLI | RMSEA (95%CI) | Model comparison | Δ CFI | Δ TLI | Δ RMSEA | Decision |
|---|---|---|---|---|---|---|---|---|---|
| M1: Configural invariance | 2.75 | .976 | .972 | .081 (.075-.087) | - | - | - | - | - |
| M2: Metric invariance | 3.19 | .968 | .965 | .090 (.085-.096) | M1 | -.008 | -.007 | .009 | Accept |
| M3: Scalar invariance | 3.69 | .955 | .957 | .100 (.095-.105) | M2 | -.013 | -.008 | .010 | Reject |
| M4: Partial scalar invariance | 3.30 | .962 | .963 | .093 (.088-.098) | M2 | -.006 | .006 | -.03 | Accept |

**Table 3. Descriptive statistics and Pearson's correlations among the measures of interest.**

| Variable | N | M | P | S | CWB-I | CWB-O | S-NAQ | AOC | OCB-I |
|---|---|---|---|---|---|---|---|---|---|
| 1. N | - | | | | | | | | |
| 2. M | .03 (-.09/.14) | - | | | | | | | |
| 3. P | .03 (.08/.15) | .46*** (.36/.55) | - | | | | | | |
| 4. S | .09 (-.03/.20) | .36*** (.26/.45) | .62*** (.54/.68) | - | | | | | |
| 5. CWB-I | -.09 (-.02/.03) | .37*** (.27/47) | .56*** (.47/.63) | .64*** (.56/.70) | - | | | | |
| 6. CWB-O | -.15** (-.27/-.04) | .37*** (.27/.47) | .59*** (.51/.66) | .54*** (.45/.62) | .75*** (.69/.80) | - | | | |
| 7. S-NAQ | .09 (-.03/.20) | .40*** (.30/.49) | .45*** (.35/.54) | .50*** (.41/.58) | .84*** (.80/.87) | .59*** (.51/.66) | - | | |
| 8. AOC | .45*** (.33/.56) | -.32*** (-.44/-.19) | -.22** (-.35/-.09) | -.14* (-.28/-.01) | -.22** (-.35/-.08) | -.39*** (-.50/-.26) | -.18* (-.32/-.04) | - | |
| 9. OCB-I | .17* (.03/.31) | -.18* (-.31/-.04) | -.46*** (-.57/-.34) | -.23*** (-.36/-.09) | -.30*** (-.43/-.16) | -.38*** (-.49/-.24) | -.14 (-.28/.01) | .30*** (.16/.42) | - |
| 10. OCB-O | .35*** (.22/.47) | -.19** (-.33/-.05) | -.43*** (-.54/-.30) | -.36*** (-.48/-.23) | -.36*** (-.48/-.13) | -.59*** (-.67/-.48) | -.26*** (-.39/-.12) | .48*** (.37/.59) | .58*** (.48/.67) |

*Note.* N = narcissism, M = machiavellianism, P = psychopathy, S = sadism, CWB-I = counterproductive behaviors at work directed at individuals, CWB-O counterproductive behaviors at work directed toward the organization, S-NAQ = short negative acts questionnaire, AOC = affective organizational commitment, OCB-I = organizational citizenship behavior directed at individuals, OCB-O = organizational citizenship behavior directed toward the organization. 95% confidence interval in parentheses.

* $p < .05$,

** $p < .01$,

*** $p < .001$

which instead displayed only a negative correlation with CWB-O. Narcissism was positively related to affective commitment and positive workplace behaviors (OCB-I and OCB-O), whereas the other DTW scales were negatively related.

Hierarchical multiple regression analyses were performed to test the contribution of the DTW scales in predicting organizational outcomes after controlling for gender and age group.

As shown in Table 4, the DTW scales along with the demographic variables explained between 27% and 40% of the variance in workplace deviance measures. Specifically, CWB-I and CWB-O were significantly predicted by psychopathy and sadism, with narcissism contributing to the prediction of CWB-O, and all four DTW scales significantly predicted S-NAQ.

The results of the hierarchical regressions with the positive workplace behaviors as dependent variables are shown in Table 5. The predictors explained between 24% and 33% of the variance in the measures of positive workplace behaviors. Narcissism and Machiavellianism significantly predicted AOC scores, and narcissism and psychopathy predicted organizational citizenship behaviors, both directed at individuals (OCB-I) toward the organization (OCB-O).

## Discussion

The aim of the present study was to contribute to the growing literature on personality traits at work by introducing and validating a version translated into Italian of the Dark Tetrad at Work scale, an instrument specifically designed to assess four socially aversive personality traits in the work context [22].

The results of the confirmatory factor analysis demonstrated that the theoretical four-factor model provided the best fit to the data, indicating good psychometric properties. Moreover, configural invariance, as well as full metric and partial scale invariance, were established across Italian and English versions, allowing cross-cultural comparison. Finally, the correlation

**Table 4. Hierarchical multiple regression analysis summary for the DTW scales predicting workplace deviance measures.**

| Predictors | CWB-I | CWB-O | S-NAQ |
|---|---|---|---|
| Gender | -.01 | -.02 | .06 |
| Age | .04 | -.13** | .03 |
| Narcissism | -.07 | -.14** | .13* |
| Machiavellianism | .07 | .07 | .20*** |
| Psychopathy | .26*** | .41*** | .18* |
| Sadism | .42*** | .21*** | .25*** |
| Summary Statistics | | | |
| Model $F$ | 30.79*** | 31.74*** | 17.93*** |
| Adjusted $R^2$ | .40 | .40 | .27 |

*Note.* Results (in standardized betas) of hierarchical multiple regression anlyses. CWB-I = counterproductive behaviors at work directed at individuals, CWB-O counterproductive behaviors at work directed toward the organization, S-NAQ = short negative acts questionnaire.

* $p < .05$,

** $p < .01$,

*** $p < .001$

analysis demonstrated the convergent and discriminant validity of the Italian version of the DTW. In particular, it showed significant correlations with measures of workplace deviance and positive workplace behaviors. These results were supported by the hierarchical regression analyses, which confirmed the predictive power of the DTW scale. The dark tetrad traits were found to be robust predictors, explaining a substantial proportion of variance in both negative and positive workplace behaviors measures.

Overall, the patterns of associations observed in this study are consistent with the results of previous studies using either the original English version [22] or the Spanish translation of the scale [23, 24]. Machiavellianism, psychopathy, and sadism showed positive associations with negative workplace behaviors and negative associations with positive workplace behaviors. Both psychopathy and sadism were significant unique predictors of the workplace deviant behaviors included in the study. This finding supports the hypothesis that everyday

**Table 5. Hierarchical multiple regression analysis summary for the DTW scales predicting positive workplace behaviors.**

| Predictors | AOC | OCB-I | OCB-O |
|---|---|---|---|
| Gender | -.05 | .01 | .08 |
| Age group | -.03 | -.02 | .15* |
| Narcissism | .45*** | .21** | .35*** |
| Machiavellianism | -.27*** | -.01 | .02 |
| Psychopathy | -.11 | -.48*** | -.41*** |
| Sadism | -.02 | -.01 | -.02 |
| Summary Statistics | | | |
| Model $F$ | 13.01*** | 10.73*** | 15.91*** |
| Adjusted $R^2$ | .28 | .24 | .33 |

*Note.* Results (in standardized betas) of hierarchical multiple regression anlyses. AOC = affective organizational commitment, OCB-I = organizational citizenship behavior directed at individuals, OCB-O = organizational citizenship behavior directed toward the organization.

* $p < .05$,

** $p < .01$,

*** $p < .001$

sadism is a separate trait from psychopathy and warrants inclusion in the Dark Tetrad [11, 13]. Narcissism, on the other hand, showed a more complex and nuanced pattern, being positively associated with positive workplace behaviors, negatively associated with counterproductive behaviors toward the organization, and positively associated with workplace bullying. This finding may reflect the already underlined multidimensional nature of narcissism, which comprises different facets representing different aspects of narcissism [42]. Furthermore, these facets also differ in their association with adaptive behaviors [43]. The leadership/authority facet is associated with positive outcomes, such as increased social support and reduced psychological distress, and is considered the healthier aspect of narcissism [16, 42, 44, 45]. According to a meta-analysis, this facet has shown a negative association with counterproductive behaviors in the workplace [46]. In contrast, the grandiose/exhibitionism and especially entitlement/exploitativeness facets represent the maladaptive aspects of narcissism. The former is linked to self-absorption and lack of humility, while the latter is associated with interpersonal difficulties and counterproductive behavior at work [43, 46]. It has been argued that the narcissism items of the DTW scale focus mainly on the leadership/authority and grandiose/exhibitionism facets, rather than on the entitlement/exploitativeness dimension [24], which could explain the negative correlations with measures of counterproductive workplace behaviors and the positive ones with measures of desirable workplace behaviors. Nevertheless, consistent with other studies [24, 47], regression analyses revealed "the dark side" of narcissism through its significant positive association with workplace bullying as measured by the S-NAC.

A strength of this study is the provision of a reliable and context-specific instrument for investigating dark personality traits in the Italian workforce with robust psychometric properties, which are also supported by testing cross-language measurement invariance. Such measurement invariance is crucial to facilitate cross-language and cross-cultural comparisons and to gain a more comprehensive understanding of how dark personality traits operate in different linguistic and cultural contexts. In addition, preliminary normative scores were also presented.

Some limitations of this study must also be acknowledged. First, the cross-sectional nature of the study does not allow for inferences about causal or temporal directions. Second, the study samples were drawn from an online survey platform, potentially limiting the generalizability of the findings to broader populations. Finally, it is essential to acknowledge that employing self-report data introduces potential sources of bias, including social desirability bias and common method bias. Consequently, caution is advised when interpreting the findings, particularly those related to narcissism, as individuals scoring high on this trait are known to exhibit heightened sensitivity to social desirability concerns [22, 48].

Future research could incorporate multi-method assessments, such as evaluations from external judges, to mitigate these potential biases.

## Conclusion

In conclusion, this study provides robust support for the factorial validity, reliability, and criterion validity of the Italian version of the DTW scale. These results are consistent with previous research [22–24]. Moreover, the cross-language measurement invariance of the scale was assessed, implying that the DTW scale effectively captures the same underlying constructs in both Italian and English-speaking populations.

With its specific focus on the workplace context, the Dark Tetrad at Work scale provides researchers with a means to assess the prevalence of dark personality traits among employees. This could help shed light on their influence on workplace behavior and examine their

relationship to job performance, job satisfaction, organizational commitment, and other relevant outcomes.

From a practical perspective, this tool could help organizations develop and implement more effective interventions to mitigate the negative impact of dark personality traits in the workplace. For example, by assessing the prevalence of employees who are more likely to engage in harmful and toxic behaviors in the workplace, organizations can proactively address potential problems before they escalate, promoting a healthier workplace culture. In addition, assessing Dark Tetrad traits in the workplace could help make informed decisions in areas such as new talent acquisition, leadership development and succession planning.

In summary, assessing Dark Tetrad traits in the workplace is an important step in developing a healthy and positive work environment that is critical to employee engagement, retention, and overall organizational performance.

## Supporting information

**S1 Table. Items, descriptive statistics, and factor loadings of the four-factor model of the DTW scale (Italian version).**
(DOCX)

**S2 Table. Descriptive statistics and scores distribution of the DTW scales in the Italian sample (N = 300).**
(DOCX)

## Author Contributions

**Conceptualization:** Francesco Marcatto.

**Data curation:** Francesco Marcatto.

**Formal analysis:** Francesco Marcatto, Lisa Di Blas.

**Investigation:** Francesco Marcatto, Isaiah Hipel.

**Supervision:** Donatella Ferrante, Kevin Kelloway.

**Writing – original draft:** Francesco Marcatto.

**Writing – review & editing:** Francesco Marcatto, Lisa Di Blas, Donatella Ferrante, Kevin Kelloway.

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
