## [Decision Letter · Decision Letter 0]

6 Dec 2023

PONE-D-23-35357Validation of the Italian version of the Dark Tetrad at Work scale: Psychometric properties and measurement invariance across languagesPLOS ONE

Dear Dr. Marcatto,

Thank you for submitting your manuscript to PLOS ONE. After careful consideration, we feel that it has merit but does not fully meet PLOS ONE’s publication criteria as it currently stands. Therefore, we invite you to submit a revised version of the manuscript that addresses the points raised during the review process.

 Dear Marcatto,

Firstly, I want to express gratitude for considering PLOS ONE as the journal to disseminate the results of your research.

To assess your manuscript, I have enlisted the participation of two expert reviewers in the field, who have conducted an independent evaluation. Based on their valuable comments, I invite you to submit a revised version of the manuscript.

Please follow the instructions included here to submit a reviewed version of your manuscript. In your response letter, address the comments from both reviewers, indicating the changes made or explaining why no changes were made and the reasons behind such decisions. Please pay special attention to comments described as "major issues."

Warm regards,

Pedro J. Ramos-Villagrasa

We look forward to receiving your revised manuscript.

Kind regards,

Pedro J. Ramos-Villagrasa, Ph.D

Academic Editor

PLOS ONE

Journal Requirements:

Reviewers' comments:

Reviewer's Responses to Questions

**Comments to the Author**

1. Is the manuscript technically sound, and do the data support the conclusions?

Reviewer #1: Yes

Reviewer #2: Partly

2. Has the statistical analysis been performed appropriately and rigorously? 

Reviewer #1: Yes

Reviewer #2: Yes

3. Have the authors made all data underlying the findings in their manuscript fully available?

Reviewer #1: Yes

Reviewer #2: Yes

4. Is the manuscript presented in an intelligible fashion and written in standard English?

Reviewer #1: Yes

Reviewer #2: Yes

5. Review Comments to the Author

Reviewer #1: The present study sought to adapt the Dark Tetrad at Work scale to Italian and compare its results with English-speaking people using a measurement invariance model. Overall, it is a well-written article with adequate analysis and theoretical background. I have some major and minor concerns described below.

Major

- The abstract is not very informative about the results found. What do the authors mean by “The associations were consistent with previous studies”? Please ensure that all necessary and detailed information for understanding the article is provided here.

- The data analysis process, although detailed, needs to provide information on why three confirmatory models were tested instead of three exploratory models. Furthermore, why are the authors using Spearman’s correlation with hierarchical regressions? If their data is non-normal, I strongly suggest they use non-parametric analysis for the whole manuscript (although, with their sample size, it would be okay to perform Pearson’s correlation with bootstraps).

Minor

- I suggest the authors to re-think their title. Presently, it is too long and repeats unnecessary information. For example, validation is one of the processes of psychometric properties. Furthermore, they mention measurement invariance across languages, which does not inform the reader about which languages were assessed. I would suggest keeping that information in the abstract only.

- I suggest removing keywords that already appear in the title, e.g., dark tetrad at work, and using keywords that maximize the reach of the manuscript once published.

- Please include more information about participants in the abstract, ensuring a complete understanding of what was performed in the article.

- In the introduction, lines 41 and 42, the authors refer to psychopathy and sadism as subclinical, but narcissism is also displayed in its subclinical form in the Dark Tetrad. Furthermore, reference 9 (lines 55-57), used to define everyday sadism, does not mention schadenfreude as a part of sadistic behavior. They may overlap traits in some respects but do not necessarily refer to the same trait. If the authors used a specific reference to make this point, please include said reference.

- The inclusion of a new meta-analysis that was performed on the topic could help the authors to discuss further the results found: Bonfá-Araujo, B., Lima-Costa, A. R., Hauck-Filho, N., & Jonason, P. K. (2022). Considering sadism in the shadow of the Dark Triad traits: A meta-analytic review of the Dark Tetrad. Personality and Individual Differences, 197, 111767 and Kowalski, C. M., Di Pierro, R., Plouffe, R. A., Rogoza, R., & Saklofske, D. H. (2020). Enthusiastic acts of evil: The Assessment of Sadistic Personality in Polish and Italian populations. Journal of Personality Assessment, 102(6), 770-780.

Reviewer #2: The paper focuses on the Italian translation and validation of the DTW scale. The paper is well-written. However, it aligns more closely with a technical report than a traditional scientific article due to its simplistic approach to complex topics in the theoretical introduction and discussion sections. I understand that the central aspect of the paper is the adaptation process, yet an expanded introduction and discussion on the empirical and theoretical links between dark tetrad personality traits and work outcomes would enhance its depth and academic value.

For instance, the paper briefly touches on the positive correlation between narcissism and organizational citizenship behaviors (OCB), attributing it exclusively to narcissists' self-promotion tendencies. This explanation appears overly simplistic, particularly in light of extensive research that illustrates a more complex relationship between narcissism and various work attitudes. Studies such as those by Papageorgiou et al. (2019) and Szabó et al. (2021) provide evidence of this nuanced relationship. These studies have found positive associations between narcissism and OCB, and non-significant/negative relationships between narcissism and CWB. The introduction of this paper suggests a straightforward positive correlation between dark personality traits and Counterproductive Work Behavior (CWB), as well as a negative correlation with Organizational Citizenship Behaviors (OCB) and organizational commitment. However, this perspective does not align with the more complex and varied findings emerging in recent literature. Furthermore, the paper's discussion asserts a positive association between narcissism and workplace bullying. This claim, however, is contradicted by the paper's own regression analysis findings, which show narcissism as a negative predictor of CWB-O and only a weak predictor of CWB-I. This inconsistency warrants further clarification to ensure the accuracy of the paper's conclusions.

The methods section is clear and well-articulated. However, the inclusion of the English sample in the study does not evidently contribute additional value. Given the notable differences in demographic composition and the absence of outcome measures for the English version, it might be more coherent to exclude this sample from the analysis. The original DTW study and the Spanish version could suffice for comparative purposes.

In conclusion, while the paper is well-structured and the methods section is strong, I would encourage the authors to work on the theoretical introduction and discussion and also consider the English sample's relevance.

References:

Papageorgiou, K. A., et al. (2019). The bright side of dark: Exploring the positive effect of narcissism on perceived stress through mental toughness. Personality and Individual Differences, 139, 116-124.

Szabó, E., et al. (2021). The importance of dark personality traits in predicting workplace outcomes. Personality and Individual Differences, 183, 1111112.

6. PLOS authors have the option to publish the peer review history of their article (what does this mean?). If published, this will include your full peer review and any attached files.

Reviewer #1: No

Reviewer #2: No

---

## [Author Response · Author response to Decision Letter 0]

12 Jan 2024

Reviewer #1: The present study sought to adapt the Dark Tetrad at Work scale to Italian and compare its results with English-speaking people using a measurement invariance model. Overall, it is a well-written article with adequate analysis and theoretical background. I have some major and minor concerns described below.

Thank you for your appreciation.

Major

- The abstract is not very informative about the results found. What do the authors mean by “The associations were consistent with previous studies”? Please ensure that all necessary and detailed information for understanding the article is provided here.

Thank you for this feedback. We completely revised the abstract so that it is now more informative about our results and the composition of the samples that participated in the study (see comment below).

- The data analysis process, although detailed, needs to provide information on why three confirmatory models were tested instead of three exploratory models. Furthermore, why are the authors using Spearman’s correlation with hierarchical regressions? If their data is non-normal, I strongly suggest they use non-parametric analysis for the whole manuscript (although, with their sample size, it would be okay to perform Pearson’s correlation with bootstraps).

Thank you for the thoughtful comment. In the 'Statistical Analysis' section we have added an explanation of why exactly we tested these three confirmatory models: 

”We followed the same procedure used in the original validation of the scale [22] and therefore tested the following three models: an unidimensional model in which all 22 items load on a single factor, a three-factor model in which psychopathy and sadism items load on the same factor (i.e., narcissism, Machiavellianism, and psychopathy-sadism), and the hypothesized four-factor model (i.e., narcissism, Machiavellianism, psychopathy, and sadism).”

We also thank you for the suggestion to use Pearson correlations, which we have taken up. We have therefore updated Table 3 accordingly (we performed Pearson correlations with 95% CI).

Minor

- I suggest the authors to re-think their title. Presently, it is too long and repeats unnecessary information. For example, validation is one of the processes of psychometric properties. Furthermore, they mention measurement invariance across languages, which does not inform the reader about which languages were assessed. I would suggest keeping that information in the abstract only.

Thank you for this suggestion. We have changed the title to: “Validation of the Italian version of the Dark Tetrad at Work scale”.

- I suggest removing keywords that already appear in the title, e.g., dark tetrad at work, and using keywords that maximize the reach of the manuscript once published.

Thank you for this suggestion. We have changed the keywords to: Dark personality; workplace personality; workplace deviance; positive workplace behaviors; measurement invariance; cross-language validation.

- Please include more information about participants in the abstract, ensuring a complete understanding of what was performed in the article.

Thank you for this feedback. As we wrote above, the abstract has been completely rewritten so that it should include more information about the participants.

- In the introduction, lines 41 and 42, the authors refer to psychopathy and sadism as subclinical, but narcissism is also displayed in its subclinical form in the Dark Tetrad. Furthermore, reference 9 (lines 55-57), used to define everyday sadism, does not mention schadenfreude as a part of sadistic behavior. They may overlap traits in some respects but do not necessarily refer to the same trait. If the authors used a specific reference to make this point, please include said reference.

Thank you for bringing this inconsistencies to our attention. We refer now to narcissism as subclinical too (in the Introduction section: “The four traits that make up the Dark Tetrad are (subclinical) narcissism, Machiavellianism, (subclinical) psychopathy, and (subclinical) sadism. Narcissism is characterized by a grandiose self-view, fantasies of control, success, and admiration, and a sense of entitlement [2]. Individuals high in subclinical narcissism often exaggerate their accomplishments, block criticism, and refuse to compromise [3].”). Moreover, we have corrected the description of everyday sadism, without mentioning schadenfreude and with a different reference: “Subclinical sadism can be viewed as the enjoyment of cruelty in many everyday activities, such as bullying or intentionally humiliating another person (direct sadism), and taking pleasure in watching violent films or hurting people in video games (vicarious sadism) [9]. The construct of sadism was met with skepticism in the early Dark Tetrad literature: its similarity to psychopathy, as both traits involve a proneness to aggression [10], raised doubts on its unique contribution beyond the established traits of the original Dark Triad (narcissism, Machiavellianism, and psychopathy) [11]. Despite possible overlaps with psychopathy, the enjoyment of cruelty is a unique feature of everyday sadism and distinguishes it from other antisocial traits [12]. Indeed, extensive research has shown that everyday sadism consistently predicts various deviant behaviors and outcomes beyond psychopathy and the other socially aversive personality traits [11; 13; 14]”

- The inclusion of a new meta-analysis that was performed on the topic could help the authors to discuss further the results found: Bonfá-Araujo, B., Lima-Costa, A. R., Hauck-Filho, N., & Jonason, P. K. (2022). Considering sadism in the shadow of the Dark Triad traits: A meta-analytic review of the Dark Tetrad. Personality and Individual Differences, 197, 111767 and Kowalski, C. M., Di Pierro, R., Plouffe, R. A., Rogoza, R., & Saklofske, D. H. (2020). Enthusiastic acts of evil: The Assessment of Sadistic Personality in Polish and Italian populations. Journal of Personality Assessment, 102(6), 770-780.

This feedback is very appreciated. We have substantially updated the literature and rewritten or added considerable portions of the text based on the references suggested by both reviewers.

Reviewer #2: The paper focuses on the Italian translation and validation of the DTW scale. The paper is well-written. However, it aligns more closely with a technical report than a traditional scientific article due to its simplistic approach to complex topics in the theoretical introduction and discussion sections. I understand that the central aspect of the paper is the adaptation process, yet an expanded introduction and discussion on the empirical and theoretical links between dark tetrad personality traits and work outcomes would enhance its depth and academic value.

For instance, the paper briefly touches on the positive correlation between narcissism and organizational citizenship behaviors (OCB), attributing it exclusively to narcissists' self-promotion tendencies. This explanation appears overly simplistic, particularly in light of extensive research that illustrates a more complex relationship between narcissism and various work attitudes. Studies such as those by Papageorgiou et al. (2019) and Szabó et al. (2021) provide evidence of this nuanced relationship. These studies have found positive associations between narcissism and OCB, and non-significant/negative relationships between narcissism and CWB. The introduction of this paper suggests a straightforward positive correlation between dark personality traits and Counterproductive Work Behavior (CWB), as well as a negative correlation with Organizational Citizenship Behaviors (OCB) and organizational commitment. However, this perspective does not align with the more complex and varied findings emerging in recent literature. Furthermore, the paper's discussion asserts a positive association between narcissism and workplace bullying. This claim, however, is contradicted by the paper's own regression analysis findings, which show narcissism as a negative predictor of CWB-O and only a weak predictor of CWB-I. This inconsistency warrants further clarification to ensure the accuracy of the paper's conclusions.

Thank you for the thoughtful comment. The main aim of this paper is to present the Italian version of the DTW scale and describe its psychometric properties. However, we agree that both the theoretical introduction and the discussion section could be improved. Therefore, we have revised both sections, taking into account the references suggested by both reviewers:

“The Dark Tetrad in the workplace

The effects of the Dark Tetrad traits can be profound and harmful, both to the individuals who exhibit these traits and to the people with whom they interact [15; 16]. In the workplace, the presence of individuals with Dark Tetrad personality traits can have several negative consequences. People with high levels of Machiavellianism and narcissism may engage in manipulative tactics to achieve their goals, often at the expense of others, and their lack of concern for others well-being may lead to exploitative and harmful behaviors [17]. For example, Machiavellianism has been identified as a significant predictor of employees' willingness to engage in corruptive behavior [18]. Similarly, narcissism has been found to be a positive predictor of the adoption of bullying tactics at the workplace [19]. High levels of psychopathy are associated with antisocial behaviors such as impulsivity, aggression, and disregard for social norms and rules. This can lead to legal and interpersonal problems, as well as difficulties in adhering to social and occupational norms [3]. In fact, people with high levels of subclinical psychopathy are more likely to intentionally engage in behaviors aimed at harming their organization [18; 20]. Lastly, people with high levels of subclinical sadism derive pleasure from others’ suffering and are therefore prone to workplace bullying, such as shaming others and openly attacking their dignity and self-esteem, especially in a public context [19].In summary, the manipulative, exploitative, and antagonistic behaviors associated with the Dark Tetrad traits can create a toxic work environment and negatively impact the overall productivity, morale, and employee well-being [21].”

In the Discussion section: “Overall, the patterns of associations observed in this study are consistent with the results of previous studies using either the original English version [22] or the Spanish translation of the scale [23; 24]. Machiavellianism, psychopathy, and sadism showed positive associations with negative workplace behaviors and negative associations with positive workplace behaviors. Both psychopathy and sadism were significant unique predictors of the workplace deviant behaviors included in the study. This finding supports the hypothesis that everyday sadism is a separate trait from psychopathy and warrants inclusion in the Dark Tetrad [11; 13]. Narcissism, on the other hand, showed a more complex and nuanced pattern, being positively associated with positive workplace behaviors, negatively associated with counterproductive behaviors toward the organization, and positively associated with workplace bullying. This finding may reflect the already underlined multidimensional nature of narcissism, which comprises different facets representing different aspects of narcissism [42]. Furthermore, these facets also differ in their association with adaptive behaviors [43]. The leadership/authority facet is associated with positive outcomes, such as increased social support and reduced psychological distress, and is considered the healthier aspect of narcissism [16; 42; 44; 45]. According to a meta-analysis, this facet has shown a negative association with counterproductive behaviors in the workplace [46]. In contrast, the grandiose/exhibitionism and especially entitlement/exploitativeness facets represent the maladaptive aspects of narcissism. The former is linked to self-absorption and lack of humility, while the latter is associated with interpersonal difficulties and counterproductive behavior at work [43; 46]. It has been argued that the narcissism items of the DTW scale focus mainly on the leadership/authority and grandiose/exhibitionism facets, rather than on the entitlement/exploitativeness dimension [24], which could explain the negative correlations with measures of counterproductive workplace behaviors and the positive ones with measures of desirable workplace behaviors. Nevertheless, consistent with other studies [24; 47], regression analyses revealed “the dark side” of narcissism through its significant positive association with workplace bullying as measured by the S-NAC.”

We added the following references in the References list:

• Ackerman RA, Witt EA, Donnellan MB, Trzesniewski KH, Robins RW, Kashy DA. What does the narcissistic personality inventory really measure? Assessment. 2011;18(1):67-87.

• Back MD, Schmukle SC, Egloff B. Why are narcissists so charming at first sight? Decoding the narcissism–popularity link at zero acquaintance. J Pers Soc Psychol. 2010;98(1):132.

• Bonfá-Araujo B, Lima-Costa AR, Hauck-Filho N, Jonason PK. Considering sadism in the shadow of the Dark Triad traits: A meta-analytic review of the Dark Tetrad. Pers Individ Dif. 2022;197:111767.

• Buckels EE, Jones DN, Paulhus DL. Behavioral confirmation of everyday sadism. Psychol Sci. 2013;24(11):2201-2209.

• Emmons RA. Narcissism: Theory and measurement. J Pers Soc Psychol. 1987;52:11-17.

• Fernández-del-Río E, Ramos-Villagrasa PJ, Escartín J. The incremental effect of Dark personality over the Big Five in workplace bullying: Evidence from perpetrators and targets. Pers Individ Dif. 2021;168:110291.

• Grijalva E, Newman DA. Narcissism and counterproductive work behavior (CWB): Meta‐analysis and consideration of collectivist culture, Big Five personality, and narcissism's facet structure. Appl Psychol. 2015;64(1):93-126.

• Johnson LK, Plouffe RA, Saklofske DH. Subclinical sadism and the Dark Triad: Should there be a Dark Tetrad? J Individ Differ. 2019;40:127–133.

• Kowalski CM, Di Pierro R, Plouffe RA, Rogoza R, Saklofske DH. Enthusiastic acts of evil: The Assessment of Sadistic Personality in Polish and Italian populations. J Pers Assess. 2020;102(6):770-780.

• Papageorgiou KA, Gianniou FM, Wilson P, Moneta GB, Bilello D, Clough PJ. The bright side of dark: Exploring the positive effect of narcissism on perceived stress through mental toughness. Pers Individ Dif. 2019;139:116-124.

• Paulhus DL, Jones DN. Measures of dark personalities. In: Measures of Personality and Social Psychological Constructs. Academic Press; 2015. p. 562-594.

• Sassenrath C, Keller J, Stöckle D, Kesberg R, Nielsen YA, Pfattheicher S. I like it because it hurts you: On the association of everyday sadism, sadistic pleasure, and victim blaming. J Pers Soc Psychol. 2023. Advance online publication.

• Scherer KT, Baysinger M, Zolynsky D, LeBreton JM. Predicting counterproductive work behaviors with sub-clinical psychopathy: Beyond the Five Factor Model of personality. Pers Individ Dif. 2013;55(3):300-305.

• Szabó ZP, Simon E, Czibor A, Restás P, Bereczkei T. The importance of dark personality traits in predicting workplace outcomes. Pers Individ Dif. 2021;183:111112.

• Tokarev A, Phillips AR, Hughes DJ, Irwing P. Leader dark traits, workplace bullying, and employee depression: Exploring mediation and the role of the dark core. J Abnorm Psychol. 2017;126(7):911.

The methods section is clear and well-articulated. However, the inclusion of the English sample in the study does not evidently contribute additional value. Given the notable differences in demographic composition and the absence of outcome measures for the English version, it might be more coherent to exclude this sample from the analysis. The original DTW study and the Spanish version could suffice for comparative purposes.

In conclusion, while the paper is well-structured and the methods section is strong, I would encourage the authors to work on the theoretical introduction and discussion and also consider the English sample's relevance.

References:

Papageorgiou, K. A., et al. (2019). The bright side of dark: Exploring the positive effect of narcissism on perceived stress through mental toughness. Personality and Individual Differences, 139, 116-124.

Szabó, E., et al. (2021). The importance of dark personality traits in predicting workplace outcomes. Personality and Individual Differences, 183, 1111112.

Thank you for this comment. The English sample has been used only for conducting the measurement invariance (across languages) analysis. In this version, we have attempted to provide clearer details regarding the utilization of both samples in the 'Measures and Procedure' section as follows: “The Italian-speaking sample was asked to complete a questionnaire that included the Italian version of the DTW, along with additional organizational measures of workplace deviance and positive workplace behaviors, in order to verify the criterion-related validity of the Italian version. Conversely, the English-speaking sample received a questionnaire presenting only the English version of the DTW, to test its cross-language measurement invariance.”

---

## [Editor Report · Decision Letter 1]

26 Jan 2024

PONE-D-23-35357R1Validation of the Italian version of the Dark Tetrad at Work scalePLOS ONE

Dear Dr. Marcatto,

Thank you for submitting your manuscript to PLOS ONE. I want to begin by expressing appreciation for your attention to the reviewers' comments. In my opinion, the new version of the manuscript is an improvement over the original submission. However, there are three very minor details I would like you to consider before accepting the article:

1) In the abstract, you mention the reliability coefficient for three out of the four dimensions of the Dark Tetrad. It would be preferable to include all four for a more comprehensive presentation.

2) In both the abstract and the limitations section, I believe the results related to Narcissism should be approached with caution. Since everything has been measured using self-report instruments, there is a possibility that socially desirable associations may be influencing the outcomes, as indicated by previous research cited in your article.

3) On page 8, there is no need to mask the university's name.

Thank you for considering my suggestions, and congratulations for your research.

Kind regards,

Pedro J. Ramos-Villagrasa, Ph.D

Academic Editor

PLOS ONE
---

## [Author Response · Author response to Decision Letter 1]

29 Jan 2024

We have implemented all the three minor changes requested by the editor. The details of these modifications are outlined in the attached “Response to reviewers” document.

---

## [Editor Report · Decision Letter 2]

1 Feb 2024

Validation of the Italian version of the Dark Tetrad at Work scale

PONE-D-23-35357R2

Dear Dr. Marcatto,

We’re pleased to inform you that your manuscript has been judged scientifically suitable for publication and will be formally accepted for publication once it meets all outstanding technical requirements.

Kind regards,

Pedro J. Ramos-Villagrasa, Ph.D

Academic Editor

PLOS ONE

---

## [Editor Report · Acceptance letter]

14 Feb 2024

PONE-D-23-35357R2 

PLOS ONE

Dear Dr. Marcatto, 

I'm pleased to inform you that your manuscript has been deemed suitable for publication in PLOS ONE. Congratulations! Your manuscript is now being handed over to our production team.

Kind regards, 

on behalf of

Dr. Pedro J. Ramos-Villagrasa 

Academic Editor

PLOS ONE